# Comparative Methods for Quantification of Sulfate-Reducing Bacteria in Environmental and Engineered Sludge Samples

**DOI:** 10.3390/biology12070985

**Published:** 2023-07-11

**Authors:** Aracely Zambrano-Romero, Dario X. Ramirez-Villacis, Noelia Barriga-Medina, Reyes Sierra-Alvarez, Gabriel Trueba, Valeria Ochoa-Herrera, Antonio Leon-Reyes

**Affiliations:** 1Instituto de Microbiología, Universidad San Francisco de Quito USFQ, Campus Cumbayá, Diego de Robles y Vía Interoceánica, Quito 17-1200-841, Ecuador; azambranor@estud.usfq.edu.ec (A.Z.-R.); dxramirez@usfq.edu.ec (D.X.R.-V.); gtrueba@usfq.edu.ec (G.T.); 2Colegio de Ciencias e Ingeniería, Universidad San Francisco de Quito USFQ, Campus Cumbayá, Diego de Robles s/n y Vía Interoceánica, Quito 17-1200-841, Ecuador; nnbarriga@usfq.edu.ec (N.B.-M.); vochoa@usfq.edu.ec (V.O.-H.); 3Laboratorio de Biotecnología Agrícola y de Alimentos, Ingeniería en Agronomía, Colegio de Ciencias e Ingenierías, Universidad San Francisco de Quito USFQ, Campus Cumbayá, Diego de Robles y Vía Interoceánica, Quito 17-1200-841, Ecuador; 4Department of Chemical and Environmental Engineering, The University of Arizona, P.O. Box 210011, Tucson, AZ 85721-0011, USA; rsierra@arizona.edu; 5Department of Environmental Sciences and Engineering, Gillings School of Global Public Health, University of North Carolina at Chapel Hill, Chapel Hill, NC 27599-3280, USA; 6Department of Biology, University of North Carolina at Chapel Hill, Chapel Hill, NC 27599-3280, USA

**Keywords:** sulfate-reducing bacteria, sludge, culture, counts, qPCR, enumeration

## Abstract

**Simple Summary:**

Sulfate-reducing bacteria (SRB) are important microorganisms in natural ecosystems and are widely utilized in engineered processes for removing heavy metals and sulfate from acid rock drainage and other wastewater. Conventional methods for quantifying SRB are time consuming and display variable results, while molecular techniques require specialized equipment and analysis. We compared microscopic cell counting, culture, and quantitative or real-time PCR (qPCR) methods for the absolute enumeration of SRB populations in engineered and environmental sludge samples. qPCR analysis showed the best performance regarding specificity, precision, and accuracy. However, it did not work for all samples due to their complex physical, chemical, and microbiological characteristics. Using a qPCR method normalized to *dsrA* gene copies and a synthetic double-stranded DNA fragment as a calibrator could be a better solution for enumerating SRB in samples from diverse origins.

**Abstract:**

This study aimed to compare microscopic counting, culture, and quantitative or real-time PCR (qPCR) to quantify sulfate-reducing bacteria in environmental and engineered sludge samples. Four sets of primers that amplified the *dsrA* and *apsA* gene encoding the two key enzymes of the sulfate-reduction pathway were initially tested. qPCR standard curves were constructed using genomic DNA from an SRB suspension and dilutions of an enriched sulfate-reducing sludge. According to specificity and reproducibility, the DSR1F/RH3-dsr-R primer set ensured a good quantification based on *dsrA* gene amplification; however, it exhibited inconsistencies at low and high levels of SRB concentrations in environmental and sulfate-reducing sludge samples. Ultimately, we conducted a qPCR method normalized to *dsrA* gene copies, using a synthetic double-stranded DNA fragment as a calibrator. This method fulfilled all validation criteria and proved to be specific, accurate, and precise. The enumeration of metabolically active SRB populations through culture methods differed from *dsrA* gene copies but showed a plausible positive correlation. Conversely, microscopic counting had limitations due to distinguishing densely clustered organisms, impacting precision. Hence, this study proves that a qPCR-based method optimized with *dsrA* gene copies as a calibrator is a sensitive molecular tool for the absolute enumeration of SRB populations in engineered and environmental sludge samples.

## 1. Introduction

Sulfate-reducing bacteria (SRB) are anaerobic prokaryotic organisms, which are an integral part of the global sulfur and carbon cycles, and their importance is due to ecological and technical reasons. SRB belongs to a morphologically diverse and heterogeneous group of microorganisms in nature and engineered anaerobic environments such as marine sediments or swamps, industrial wastewater treatment systems, or oil and gas production facilities [1,2,3,4]. SRB are responsible for biogenic sulfide generation as part of their metabolism since they use sulfate as a terminal electron acceptor [2,5].

The application of SRB in the remediation of metal-contaminated effluents such as acid rock drainage (ARD) is of great practical and scientific interest. ARD is the result of the oxidation of sulfide minerals exposed to air and water, as well as a product of the lixiviation process in the mining industry, and they are characterized by high heavy metal content, high concentration of sulfate, and low pH values [6,7,8]. Bioremediation of ARD through sulfate-reducing microorganisms is mainly based on metal bioprecipitation [9]. Thus, SRB are widely used to remove heavy metals in efficient bioremediation processes of wastewater from the mining industry, semiconductor manufacturing, and groundwater systems [2,10,11]. SRB excretes biogenic sulfide, which reacts with metal ions to produce metallic sulfides of low solubility [12,13]. Another benefit of applying sulfate-reducing biotechnologies to remediate ARD is the decrease in acidity in treated effluents [14,15].

Research projects and technical applications nowadays require accurate assessments of SRB’s enumeration, occurrence, distribution, diversity, and community structure in various environments [6,16]. Indeed, the enumeration of metabolically active microorganisms allows researchers and operators to evaluate if the microbial community is degrading or removing pollutants, if the treatment system is functioning optimally, and if bioremediation strategies could be optimized [17,18,19]. Moreover, identifying essential functional microorganisms and exploring their metabolic capabilities contributes to scientific understanding and the development of novel bioremediation strategies [20,21].

SRB can be detected and enumerated by conventional and molecular methods [16,22]. Conventional or traditional analytical methods include culture methods, optical density measurements, and microscopic imaging enumeration techniques. These methods offer advantages since they are inexpensive and easy to implement. However, they are often time-consuming and challenging to handle in field facilities or may suffer from limitations in terms of sensitivity, repeatability, and selectivity [23,24]. Moreover, oxygen-free cameras are needed for their cultivation since these bacteria only grow in anaerobic conditions [25]. On the other hand, molecular methods provide fast and precise information about the amount and distribution of microorganisms in a specific environment [26,27]. Nevertheless, the taxa identified through molecular methods differ from the viable bacteria quantified using culture-dependent methods. Despite that, cultivation-based approaches overlook the presence of viable versus non-culturable bacteria [28].

Within molecular techniques, metagenomic sequencing is a powerful method for detecting diverse sulfate-reducing bacteria (SRB) [16,29]. This technique provides relative cell counts and requires specialized sequencing equipment and bioinformatic analysis. Therefore, its application is justified in studies seeking to understand the composition and structure of complex microbial communities [16,30]. In studies where precise absolute cell counts are necessary, such as those involving cell concentration assessments or process performance evaluations, alternative, more direct, and specific methods, such as cell counting, should be employed [3,31,32].

Quantitative or real-time PCR (qPCR) has also been reported as an SRB enumeration molecular method [32,33,34]. Two functional and highly conserved genes, *apsBA* and *dsrAB*, are widely used as phylogenetic markers for identifying and quantifying SRB. These genes encode two key enzymes in the sulfate-reduction pathway, which are adenosine-5′-phospho-sulfate reductase (APS), and dissimilatory sulfite reductase (DSR) [2]. APS catalyzes the reduction of adenosine-5′-phosphosulfate to sulfite (SO_3_^2−^), the initial step in reducing sulfate to sulfide that provides the reducing power necessary for subsequent steps in the pathway. Moreover, DSR is specific to the dissimilatory pathway of sulfate reduction and catalyzes the reduction of sulfite to sulfide (S^2−^), the final step in the pathway [35]. Some studies have validated using these two genes as amplification targets by qPCR for the quantification of heterogenic SRB and phylogenetic analysis [6,32]. Ben-Dov et al. [36] designed four sets of primers from conserved regions of multiple alignments of *apsA* and *dsrA* genes and developed a method based on qPCR to quantify SRB in complex environmental and industrial water samples. However, the results of that research displayed high variability when used in soil or sludge samples [36].

As mentioned, qPCR methods for the absolute enumeration of SRB provide a fast way to overcome problems associated with microscopic and culture-dependent methods [2,2,37]. However, even successful molecular techniques require validation or verification before application in each sample, depending on their type (matrix) and complex soil/sludge type from many origins [16,36,38]. Sludge and soil samples from engineered or natural environments are characterized by their complex physical, chemical, and microbiological composition. These samples exhibit various physical properties, including varying particle sizes, textures, and structures, impacting water-holding capacity, porosity, permeability, and nutrient availability. Moreover, they consist of a combination of organic matter (such as humic substances, plant residues, debris, and microbial biomass), inorganic compounds (including minerals, metals, and ions), and various pollutants (e.g., heavy metals, pesticides, and industrial contaminants). The presence of these chemical components varies depending on the sample’s origin [39,40,41,42]. Additionally, sludge and soil samples host diverse microorganisms, including bacteria, fungi, archaea, viruses, and protozoa, forming intricate and interactive microbial communities. The composition and activity of these microbial populations vary across different environments, posing challenges in analyzing a specific group of microorganisms such as SRB [38,43,44].

In light of the above, our study compared various analytical methods for quantifying sulfate-reducing bacteria in environmental and engineered sludge samples. Initially, we assessed counting, culturing, and qPCR analysis using four sets of primers for *dsrA* and *apsA* genes previously designed and tested by Ben-Dov et al. [36]. We employed genomic DNA extracted from bacterial suspensions and dilutions of enriched sulfate-reducing sludge to establish standard curves based on complex SRB populations. Bacterial counts of SRB cultivated under anaerobic conditions from six samples were also determined. Furthermore, we compared SRB counting methodologies using Neubauer chamber counting (total), culture plating (viable), and qPCR. Subsequently, we carried out a qPCR analysis using the primers DSR1F/RH3-dsr-R and a synthetic DNA standard as an accurate method for estimating the number of copies of the *dsrA* gene in samples coming from a range of origins, including bacterial suspensions, engineered or enriched sludges, a sludge from a sulfate-reducing bioreactor, and an environmental sludge from a lagoon.

Hence, we introduce a reliable qPCR methodology for quantifying SBR in complex sludge samples within a bioremediation approach to treat mining effluents. Our study goes beyond the mere comparison and execution of existing enumeration techniques by introducing and validating a culture-independent approach that effectively overcomes some of the limitations of current methods. By offering valuable insights, our research contributes to the advancement of the accurate enumeration of diverse and dissimilatory SRB populations in both natural and engineered sludge samples, thereby enhancing our understanding of their potential biotechnological applications.

## 2. Materials and Methods

### 2.1. Basal Mineral Medium (BMM)

The basal mineral medium contained: 280 mg L^−1^ NH_4_Cl; 195 mg L^−1^ KH_2_PO_4_; 49 mg L^−1^ MgSO_4_; 10 mg L^−1^ CaCl_2_; 3000 mg L^−1^ NaHCO_3_; 10 mg L^−1^ yeast extract; 2900 mg L^−1^ Na_2_SO_4_; 5300 mg L^−1^ CH_3_COONa and 1 mL L^−1^ of a solution of trace elements. The trace element solution contained: 50 mg L^−1^ H_3_BO_3_; 2000 mg L^−1^ FeCl_2_∙4H_2_O; 50 mg L^−1^ ZnCl_2_; 32 mg L^−1^ MnCl_2_; 50 mg L^−1^ (NH_4_)_6_ Mo_7_O_24_∙4H_2_O; 50 mg L^−1^ AlCl_3_; 2000 mg L^−1^ CoCl_2_∙6H_2_O; 50 mg L^−1^ NiCl_2_∙6H_2_O; 44 mg L^−1^ CuSO_4_∙5H_2_O; 100 mg L^−1^ NaSeO_3_∙5H_2_O; 1000 mg L^−1^ EDTA; 200 mg L^−1^ resazurin and 1 mL L^−1^ of hydrochloric acid (36%) [45].

### 2.2. Samples

#### 2.2.1. Environmental Sample

Anaerobic sludge (Figure 1a, I) was collected from sediments at the bottom, approximately 1.2 m deep, of a lagoon at Universidad San Francisco de Quito, USFQ, Quito-Ecuador (0°11′49″ S 78°26′09″ W). The sludge was stored at 4 °C. The content of total suspended solids (TSS) and volatile suspended solids (VSS) in the sludge were 51.7% and 5.9%, respectively. The maximum specific sulfidogenic activity was 4337 mg S^2−^ kg^−1^ VSS d^−1^.

#### 2.2.2. Engineered Samples

Sludge sample from a sulfate-reducing (SR) bioreactor. A sludge sample was obtained from an SR lab-scale bioreactor, initially inoculated with anaerobic sludge from a lagoon. This bioreactor was used for the bioprecipitation of copper and zinc (Figure 1a, II), whose sulfate-reducing activity had been monitored for two years [30,46]. The Shannon index (alpha-diversity) was 2.72 [30]. The sample was collected in sterile plastic recipients, stored at 4 °C, and used for later comparison of enumeration methods.Enriched sulfate-reducing (SR) sludge. This sample was obtained from anaerobic sludge (Figure 1a, III), the same as described in the previous section, which was enriched in a selective liquid culture medium composed of BMM supplemented with sodium acetate equivalent to 2.5 g of chemical oxygen demand (COD) L^−1^ as organic substrate and 2.0 g SO_4_^2−^ L^−1^ as sodium sulfate (1.25:1, *w/w* ratio). Both reagents were provided by JT Baker Chemical Company (Phillipsburg, NJ, USA). Enrichment was conducted under anaerobic conditions in triplicate using a sterile liquid culture medium containing 10% *w/v* of anaerobic sludge, and the culture medium was not replaced during incubation time (40 days). The cultures were cultivated in 160 mL sterile glass serum bottles that were hermetically sealed with butyl rubber stoppers and aluminum crimp seals. The bottle headspace was flushed with nitrogen gas, and all bottles were incubated for 45 days in darkness in a climate-controlled chamber at 30 ± 2 °C. The enriched SR sludge was analyzed immediately by conventional methods. Likewise, genomic DNA extracts from serial dilutions of enriched SR sludge were obtained and stored at −80 °C. The Shannon index was 3.64 [30]. Following the previous procedure, two independent SR sludge samples (standard and control) were prepared to compare enumeration methods later.

### 2.3. Preparation of SRB Suspension

SR sludge was pour-plated in a culture medium composed of BMM supplemented with sodium acetate 2.5 g COD L^−1^ as organic substrate and 2.0 g SO_4_^2−^ L^−1^ (1.25:1, *w/w* ratio), sodium acetate and sodium sulfate were provided by JT Baker Chemical Company (Phillipsburg, NJ, USA); 1.5% agar *w/v* (BactoTM Agar, Difco Laboratories, Bordeaux, France) was added. Immediately after the incubation time (28–30 days) at 30 °C under anaerobic conditions [25], isolated colonies were picked off and placed into bottles containing sterile BMM supplemented with acetate 2.5 g COD L^−1^ as organic substrate and 2.0 g SO_4_^2−^ L^−1^ as sodium sulfate (1.25:1, *w/w* ratio). After another incubation period, according to the described conditions, SRB suspension (Figure 1a, IV) was re-cultured for plate count and enumerated in a Neubauer counting chamber (BOECO, Hamburg, Germany). Then, genomic DNA from the SRB suspension was extracted and kept at −80 °C until the beginning of the molecular tests. Following the previous procedure, two independent SRB suspensions (standard and control) were prepared to compare enumeration methods later.

### 2.4. Cell Enumeration Using Neubauer Chamber

Total SRB in the enriched SR sludge and the SRB suspension were enumerated using a Neubauer counting chamber (Figure 1b) during their exponential growth stage [23]. While the calibrators (SRB suspension and enriched SR sludge) were cultured for colony counting, they were diluted in a sterile saline solution (0.9% NaCl, *w/v*) for cell counting using a Neubauer chamber with 0.1 mm depth. Bacteria morphology was determined because of the SRB cells’ characteristic black or dark brown color [3]. The number of total SRB per milliliter (mL) was calculated following the instructions provided by the supplier. Total SRB counts included viable and non-viable cells. Each sample was analyzed in triplicate.

### 2.5. Plating Culture and Colony Counting (CFU Method)

Simultaneously to Neubauer chamber cell enumeration and during their exponential growth stage, both the SRB suspension and SR sludge were plate cultured (Figure 1b). A sample aliquot (1 mL) was serially diluted in 9 mL of sterile BMM supplemented with acetate and sulfate at a 1.25:1 *w/w* ratio to quantify viable SRB. An aliquot of 100 µL of raw sample and each corresponding dilution (10^−1^ to 10^−9^) were pour-plated in three replicates. Culture medium was supplemented with 1.5% agar *w/v* (BactoTM Agar, Difco Laboratories, Bordeaux, France), cooled to 45 °C, and a volume of 20 to 25 mL was dispensed per plate. The agar plates were incubated in an anaerobic chamber [25] (N_2_/CO_2_ atmosphere, 80/20 *v/v*) at 30 ± 0.5 °C for 28–30 days, and then the colonies were counted. The number of viable SRB, expressed as colony-forming units (CFU) per milliliter (mL), was calculated by dividing the number of colonies by the sample volume and multiplying by the dilution factor. Each sample was analyzed in triplicate.

### 2.6. DNA Extraction

DNA was extracted by standard protocols, including several purification steps for eliminating humid acids that interfere with subsequent molecular techniques [30,36,46]. A volume of 12 mL of sludge or bacterial suspension was centrifuged for 30 min at 4000× *g*. The supernatant was discarded, and the pellet was used for DNA extraction using DNeasy^®^ PowerSoil^®^ Kit (QIAGEN GmbH., Hilden, Germany) according to the protocol provided by the supplier and a previous study [27]. The purity, as the concentration of the resulting DNA preparation, was determined spectrophotometrically by the Qubit^®^ system (Thermo Fisher, Carlsbad, CA, USA). The DNA integrity of molecular weights over 2000 pairs of bases (bp) was evaluated using 2.0% agarose gels (Bioline, London, UK) with Invitrogen SYBR^®^ Safe DNA gel stain (Thermo Fisher) and TAE buffer solution (40 mol L^−1^ tris-hydroxymethyl-aminomethane, 20 × 10^−3^ mol L^−1^ acetic acid, and 1 × 10^−3^ mol L^−1^ EDTA). Electrophoresis was carried out at 80 mV for 30 min in a Gel XL EnduroTM chamber (Labnet, Edison, NJ, USA) and using 2 µL of DNA per well. Additionally, DNA of *E. coli* ATCC^®^ was extracted using an E.Z.N.A^®^ Bacterial DNA kit (Omega Bio-tek, Norcross, GA, USA), following supplier directions.

### 2.7. qPCR Analysis

qPCR assays (Figure 1b) were performed using a PrimeQ thermal cycler system (Cole-Parmer, Staffordshire, UK), based on fluorescence resonance energy transfer, using SYBR green fluorophore in a 96-well optical plate at the Laboratory of Agricultural and Food Biotechnology, Universidad San Francisco de Quito (USFQ, Quito, Ecuador). Four primer sets for qPCR amplification of *dsrA* and *apsA* genes were employed (Table A1): DSR1F and RH3-dsr-R (5′ACSCACTGGAAGCACG3′ and 5′gGTGGAGCCGTGCATGTT3′, size: 222 bp), APS7-F and RH2-aps-R (5′GGGYCTKTCCGCYATCAAYACATGA3′ and 5′ATCATGATCTGCCAgCGgCCGGA3′, 279 bp), RH1-dsr-F and RH3-dsr-R (5′GCCGTTACTGTGACCAGCC3′ and 5′gGTGGAGCCGTGCATGTT3′, 164 bp) and, RH1-aps-F and RH2-aps-R (5′CGCGAAGACCTKATCTTCGAC3′ and 5′ATCATGATCTGCCAgCGgCCGGA3′, 191 bp), where boldface represents degenerate or wobble bases, and lowercase represents bases that do not match appropriately in sequences, as described by Ben-Dov et al. [36]. These primers were designed from conserved regions of multiple alignments of *dsrA* or *apsA* sequences of SRB from industrial wastewater [36]. Invitrogen^TM^ (Grand Island, NY, USA) manufactured these four primer sets. The reaction mixture consisted of 7.5 µL of SsoFast™ EvaGreen^®^ Supermix (BioRad, Hercules, CA, USA), 150 × 10^−9^ mol L^−1^ for each forward and reverse primer, 1.5 µL of DNA template, and PCR water to a final volume of 15 µL. PCR cycles were 2 min at 50 °C, 15 min at 95 °C, followed by 40 rounds of 15 s at 95 °C [36], 1 min at 62 °C as an annealing step, and the extension step was set to 72 °C for 30 s. Melting or dissociation curves (a negative derivative of fluorescence versus temperature) were determined for nonspecific amplification products (presence or absence). The size of the PCR products was verified with SYBR^®^ Safe-stained 2.0% agarose gels. The electrophoresis conditions were 80 mV for 60 min. All runs included a blank or no-template control (Ultrapure™ Distilled Water, Invitrogen. Thermo Scientific, Walthan, MA, USA) and a negative control from a non-SRB strain (DNA of *E. coli* ATCC^®^ 25922) ran for each sample.

### 2.8. qPCR Analysis Using Viable Cells as Normalized Units

Standard curves for further improvement of the quantification method using qPCR, were generated using different concentrations of template DNA obtained from SRB suspension and serial dilutions of enriched SR sludge. The first calibrator consisted of a DNA template obtained from bacterial suspensions cultured with isolated SRB colonies (six serial dilution points in ten-fold steps). Other calibrators consisted of extracted DNA from each serial dilution of enriched SR sludge. Calibrators were normalized according to the number of total SRB cells. Standard curves were obtained by plotting each dilution point’s qPCR threshold cycle (Ct) for the DNA templates obtained by the two-different sample sources versus corresponding normalized units (viable SRB). Data were analyzed using QuanSoft^®^ software (Cole-Parmer, Staffordshire, UK). The qPCR reaction conditions were the same for each primer set. Both calibrators were run independently in triplicate, and the standard deviation was calculated for each point.

### 2.9. qPCR Analysis Using a Synthetic Standard to Enumerate dsrA Gene Copies

A double-stranded gBlocks^®^ 222 bp of dsrA gene fragment was synthesized (Integrated DNA Technologies, Coralville, IA, USA) as a qPCR standard according to the supplier’s instructions. The gBlock^®^ suspension (starting concentration: 4.39 × 10^10^ gene copies μL^−1^, equivalent to 10 ng μL^−1^) was serially diluted 10-fold from 1.0 to 1.0 × 10^−10^. Analysis was performed according to the same methodology (Section 2.7), and the resulting Ct values from each dilution gBlocks^®^ were used to construct the calibration curve. To ensure reliable quantification, by plotting the logarithm of the concentration or copy number of each standard against its Ct value, the PCR efficiency (E) was calculated using the regression equation that describes the linear relationship [31,47]. Then, the regression equation was used to calculate the dsrA gene copies in the six above samples and determine the gene ratio, which means the number of dsrA gene copies per total or viable cells [48].

### 2.10. Statistical Analysis

The effect of the method for the direct enumeration of SRBs, both in bacterial suspension and in enriched SR sludge, was determined by standard deviations calculated in SPSS Statistics software (IBM Corp., New York, NY, USA). Similarly, standard curves from two different SRB qPCR calibrators were compared through Fisher’s test (homogeneity of variances) and Student’s *t* test (slopes similarity of linear regression). The different quantification methodologies were compared by Student *t* test and analysis of variance (ANOVA), with 95% of the confidence interval, over the log_10_-transformed counts using the SPSS Statistics software. Multiple linear regression analysis was performed in R-4.3.1 software.

## 3. Results

### 3.1. SRB Enumeration Using Cell and Plate Counting Methods

To identify differences in SRB quantification by conventional methods, one SRB suspension and one enriched SR sludge sample were used for the total count of microorganisms in the Neubauer chamber and viable SRB cell enumeration by standard culture plating.

The cell density estimated using a Neubauer chamber in the SRB suspension was 2.31 × 10^6^ cells mL^−1^ (Figure 2), whereas the SRB colony count determined by culture plating was 1.79 × 10^6^ CFU mL^−1^ (Figure 2). This result showed similar results when both methods were used (*p* = 0.266). On the other hand, the bacterial concentrations determined with these two conventional methods using the enriched SR sludge sample were 7.63 × 10^6^ cells mL^−1^ (total count with Neubauer) and 8.70 × 10^5^ CFU mL^−1^ (viable cells using plate count). These results showed that almost all the cells were viable in the SRB synthetic suspension, but only 12% of the total SRB were viable in the SR sludge. These differences in outcome upon bacterial count using both techniques indicate that sample source matters for accurate determination using these two methods (*p* = 0.015).

### 3.2. Correlation of Viable SRB Count versus qPCR Data (Ct) Using an In-House Standard

Standard curves were constructed to evaluate the in-house standards’ performance and determine the best pair of primers by plotting the mean of Ct values obtained from qPCR versus log_10_ of viable SRB cells (direct counting method) from samples from SRB suspensions and synthetic sludge. These curves were generated by assuming that SRB have only one *dsrA* gene copy and one *apsA* gene copy [34]. When bacterial counts (*x*-axis) were correlated with Ct values (*y*-axis), linear regressions were obtained, and the fitness of the qPCR method and its performance was validated by the quantitative real-time PCR method proposed by Broeders et al. [49].

Figure 3a,b show the standard curves for the *dsrA* gene with sets of primers DSR1-F/RH3-dsr-R and RH1-dsr-F/RH3-dsr-R, respectively, and Figure 3c,d show the standard curves when constructed with the *apsA* gene using a set of primers APS7-F/RH2-aps-R and RH1-aps-F/RH2-aps-R, respectively. In each plot, data points represent the average of three measurements, and the standard deviations are represented with error bars. The linearity and proportional range of the molecular method developed in this study were evaluated using a linear regression analysis correlation coefficient (R^2^). During the validation of qPCR methods, R^2^ values higher than 0.98 were desirable, but that value is referential [49]. Standard curves for the *dsrA* gene (primers DSR1F/RH3-dsr-R and RH1-dsr-F/RH3-dsr-R) showed an R^2^ over 0.98 in a concentration range between 10^2^ and 10^6^ (SRB suspension), and 10^3^ to 10^7^ (enriched SR sludge) of viable SRB mL^−1^ (Figure 3a,b). On the other hand, in the same ranges, standard curves for the *apsA* gene (Figure 3c,d) showed an R^2^ between 0.94 and 0.97, which was not an acceptable parameter according to the R^2^ threshold established by Broeders and co-workers [49]. Thus, the two sets of primers that amplify the *dsrA* gene provided better linearity during qPCR standard curve construction.

### 3.3. Primers Selection Based on Accuracy from qPCR Analysis and Culture Plating

Independent samples from the SRB suspension and enriched SR sludge were analyzed to assess whether the standard curves between qPCR values (Ct) and viable SRB were accurate. The concentration of SRB in a standard control from an SRB suspension was 2.31 × 10^6^ CFU mL^−1^ (Table 1), while the concentration of SR sludge sample was 5.63 × 10^5^ CFU mL^−1^ (Table 2).

When four pairs of primers mentioned earlier were tested, SRB concentrations may be overestimated by over 100-fold, which depended on the primer set and calibrator curve used. In Table 1 and Table 2, the results show that the SRB concentrations quantified (in suspension and sludge samples) using standard curves for primer set DSR1F/RH3-dsr-R and RH1-dsr-F/RH3-dsr-R for the *dsrA* gene were the closest to theoretical value compared to those concentrations determined with standard curves for primer sets APS7-F/RH2-aps-R and RH1-aps-F/RH2-aps-R for the *apsA* gene. According to relative error (Ԑt), the most accurate enumeration of SRB using qPCR in both samples was determined using the standard curve for the DSR1F/RH3-dsr-R primer set, with a range 0.063 to 0.014 times higher than theoretical concentration.

### 3.4. Cells Enumeration by qPCR in Engineered and Environmental Samples

To test the conventional methods and the developed qPCR, we quantified the SBR in samples coming from engineered and natural environments. A total of six samples were measured by Neubauer chamber (total count), culture plating (viable count, CFU), and qPCR (DSR1F/RH3-dsr-R primer set), using both standard curves for calculation (qPCR (1): SRB suspension standard; and qPCR (2): enriched SR sludge standard).

As depicted in the corresponding figure (Figure 4), all enumeration methods yielded statistically similar bacterial counts when suspension samples were analyzed. However, we found significant differences in bacterial counts between total counts, culture plating, and qPCR methods when the sludge samples were analyzed, always giving a significantly higher bacterial count using the Neubauer chamber method. Moreover, if we compared qPCR calculation methods (1 and 2), we did not find any statistically significant differences in results obtained from independent samples using both standard curve calculations. These findings suggest that molecular analysis based on qPCR is more reproducible and specific than Neubauer chamber and culture plating counting among samples from different origins. Furthermore, as the origin’s complexity of the sample increases (environmental sludge and SR reactor sludge), the molecular method becomes less precise and reproducible with respect to total and viable counts.

### 3.5. Determination of dsrA Gene Copies

To validate the convention cell counting methods, we standardized a qPCR to quantify the number of copies of the gene *dsrA* and contrasted the results with the SBR enumeration. We constructed standard curves plotting Ct versus a number of gene copies using a synthesized double-stranded DNA gBlocks^®^ 222 bp *dsrA* gene fragment (Integrated DNA Technologies). Figure 5 shows the log_10_ concentration of serial dilutions of gBlocks^®^ plotted versus the Ct values (R^2^ = 0.997, E = 108%), evidencing good linearity of the regression equation and an efficiency between 90% and 110% [31,47,49].

After quantifying the number *dsrA* copies in the six samples, we compared the average results to the Neubauer chamber counting (total SRB cells) and the plate culture counting (viable SRB cells), and calculated the theoretical number of copies of the gene per cell (Table 3).

In all cases, total SRB cells counts were higher than viable cells (Table 3). Moreover, the counts of total cells were higher than the number of *dsrA* gene copies. Likewise, the number of viable cells was higher than the copy number of the target gene in five of the six samples (two SRB suspensions, two SR sludge samples, and a sludge sample from an SR bioreactor). Between those highlights, the sludge sample from a sulfate-reducing bioreactor had a gene ratio of 0.83. In addition, the environmental sample (anaerobic sludge of an artificial lagoon) had a gene ratio of 1.64, consistent with multiple average numbers of *dsrA* gene copies per SBR cell.

At last, even if viable cell count was higher than the number of *dsrA* copies, it showed a strong and positive correlation (Pearson’s correlation coefficient, r = 0.949, *p*-value = 0.004) with the number of *dsrA* gene copies, suggesting that both techniques may provide a consistent estimation for enumerating SRB (Figure 6a). Nevertheless, we have identified a potential bias arising from the inclusion of environmental sludge in the scatter plot. This sludge sample exhibits an atypical behavior that deviates significantly from others in magnitude and is more complex due to its origin, which impacts the correlation coefficient and the statistical significance (Figure 6b).

As shown Figure A1a,b, we did not observe any significant correlation between the counts of total SRB cells and viable cells, neither between *dsrA* gen copies and total SRB cells, respectively.

## 4. Discussion

Accurately enumerating SRB in sludge samples from various environments is critical for evaluating treatment effectiveness, optimizing strategies, identifying system failures, and advancing bioremediation research by uncovering potential metabolic capabilities. To achieve this, we conducted a comparative analysis of bacterial counts using total count, culture-dependent, and qPCR methods (Figure 1).

In this study, when the SRB concentration obtained by standard culture plating (viable, CFU) was compared with the cell density determined by the Neubauer chamber method in bacterial suspensions, there was no statistical difference, suggesting that the viable/culturable fraction and total cell count were similar (Figure 2). These results indicate that both methods were valid for SRB enumeration in bacterial suspensions.

However, the total count of SRB in enriched SR sludge was one order of magnitude higher than the concentration of colonies determined by plating (Figure 2), indicating that the concentration of total SRB was significantly higher than the concentration of viable SRB. Enriched SR sludge contains various microorganisms and mineral particles, which can potentially influence the accurate estimation of total cells when using the Neubauer chamber method [50]. Additionally, in enriched cultures, the growth of certain non-SRB microorganisms can be favored [29]. Moreover, the Neubauer chamber was enumerated under visible light illumination without contrast dyes; consequently, the total microorganisms counts could record non-SRB microorganisms [29,51].

The average relative reproducibility (standard deviation) of the viable colony counts method was ~24%, and for the Neubauer total counts method, it was ~33%. These values were comparable with results reported in other studies 7 November 2023 11:00:00 AM, showing that although culture and counting techniques for bacterial enumeration are time demanding, these methodologies remain consistent for the enumeration of microorganisms [31]. Therefore, in this study, bacterial counts for qPCR standard curves were based on values obtained from colony counts by culture plating (viable).

In constructing qPCR curves, the standard deviation of Ct values was inherent to inter-run variation, and it was a good indicator of method reproducibility/precision, as supported by Bustin et al. [52]. The relative reproducibility standard deviation (RSDr) criteria of replicates of standard curves plotted for all four pairs of primers did not exceed 10% (data now shown), validating the precision of our analytical method. RSDr should be lower than 25%, according to Broeders et al. [49].

There were no statistical differences in the slopes of standard curves obtained from two different calibrators of the same primer set (Figure 2 and Figure 3, and Table A2). Furthermore, this comparison initially evaluated the qPCR method’s applicability since results were similar using different primers [49].

The accuracy criterion based on a relative error, Ԑt, can be calculated using the SRB concentration determined by enumeration based on culture plating as the “true value” of cell density [36,49]. Consequently, accuracy was assessed by comparing the value obtained from qPCR (estimated count) and cell density determined by the culture plating value (viable count). Comparison of the value estimated using qPCR with the set of primers validated in Figure 2 shows that in some cases, SRB concentrations may be overestimated by over 100-fold from the SRB cell density value depending on the primer set used (Table 1 and Table 2). Amplification reactions using primer sets DSR1F/RH3-dsr-R and RH1-dsr-F/RH3-dsr-R of *dsrA* gene were closer to the SRB cell density values than the concentrations determined using standard curves with primer sets APS7-F/RH2-aps-R and RH1-aps-F/RH2-aps-R of the *apsA* gene (Table 1 and Table 2). This was likely because primers for the *dsrA* gene display high specificity for most SRB species, unlike primers for the *apsA* gene with poor specificity for some genera such as *Desulfacinum* and *Desulfotomaculum* [36].

The most accurate enumeration of SRB by qPCR was obtained with the DSR1F/RH3-dsr-R pair of primers (Table 1 and Table 2), with values in the range of 0.063 to 0.14 times higher than SRB cell density determined by the culture plating of both calibrators (error values ranging from 6.3 to 14.1%). In another study using genomic DNA of *Desulfovibrio vulgaris* in pure culture and with plasmids containing the *dsrA* and *apsA* genes, Ben-Dov et al. [36] developed qPCR standard curves that overestimated the number of SRBs in highly saline industrial wastewaters by 2.7- up to 10.5-fold using the DSR1F/RH3-dsr-R primer pair, which was chosen in this study as the best primer set for accurate enumeration of SRBs in samples tested.

Even though environmental samples (such as sludge) are microbially diverse, the DSR1F primer is highly specific for many species of SRB [23,36]. However, the RH3-dsr-R primer aligned with a consensus region of all SRB genera tested by Ben-Dov et al. [36], as suggested by Kondo et al. [34]. This fact also indicates that qPCR analysis using the DSR1F/RH3-dsr-R primer set does target a more prominent and heterogenic SRB population in complex sludge of environmental origin.

According to ANOVA, the two SRB suspensions samples (1 and 2) did not show significant statistical differences in SRB concentrations, independently of the quantification method (Figure 4). In both samples of enriched SR sludge, results of enumeration by Neubauer chamber (total SRB) were statistically different from those of other quantification methods, showing that enumeration by Neubauer chamber under visible light gives higher values for enumeration of SRB in sludge samples because it overestimates the bacterial concentration. Indeed, microorganisms can be associated with soil particles or biofilms, which give interferences for direct counts under a microscope [53]. Viable count and qPCR using both standard curves gave no significant differences, showing that viable count and qPCR values produce comparable results in enriched sludge samples. In contrast, when testing the environmental sludge (sample from the lagoon), the cell densities of 1.27 × 10^3^ and 6.23 × 10^3^ SRB mL^−1^ were determined by qPCR using calibrators for the calculation of SRB suspension and SR sludge values, respectively (Figure 4). Here, no significant differences were found in both samples when bacterial enumeration was performed by qPCR. However, the viable count statistically differed from the total count, showing significant differences between those two methods. qPCR analysis provided counting values between viable SRB (performed by plating) and total cells (performed using a Neubauer counting chamber). In the sludge from an SR bioreactor, the means of the counts of SRB quantified by qPCR using both calibrators were 7.48 × 10^5^ and 1.79 × 10^6^ SRB mL^−1^, respectively. There were no statistically significant differences between the samples, even when the environmental samples were tested. Also, the total and viable SRB count was statistically different in each of these samples.

To quantify SRB in sludge samples containing complex microbial communities, we plotted qPCR standard curves using genomic DNA from diverse SRB species, not unique bacterial species, such as *Desulfovibrio vulgaris* [24]. No interference from the genomic DNA of other prokaryotic microorganisms was detected, and the analytical method seemed to be specific and precise for quantifying SRB in samples from different origins [54,55,56]. After qPCR validation using standard curves and testing four sets of primers, the primers DSR1F/RH3-dsr-R based on the *dsrA* gene showed the best stability and accuracy. In addition, according to the study of Ben-Dov and coworkers [36], the primer pair DSR1F/RH3-dsr-R demonstrated high specificity for a wide range of SRB species, including various genera such as *Desulfovibrio*, *Desulfomonas*, *Desulfatibacilus*, *Desulfomicrobium*, *Desulfobacterium*, *Desulfosarcina*, *Desulfonema*, *Desulfofaba*, *Desulfomusa*, *Desulfohalobiaceae*, *Desulfotignum*, *Desulfomaculum*, *Desulfofacinum*, *Desulfonatronum*, *Desulfarculus*, *Desulfovirga*, *Desulfobulbus*, *Desulfobacter*, *Desulfococcus*, *Desulfocapsa*, *Desulfofustis*, *Desulfobacula*, *Desulfocella*, and *Desulfofospira*. In our previous research [30], we sequenced the V3 and V4 regions of the 16S rRNA genes in enriched sulfate-reducing sludge and sludge from a sulfate-reducing bioreactor, which allowed taxonomic identification of various groups, including *Desulfobacteraceae*, *Desulfuromonadales*, *Desulfotomaculum*, *Desulfovibrio*, *Desulfobulbaceae*, *Desulfobacca*, *Desulfocurvus*, *Desulfoluna*, *Desulfovibrionales*, and *Desulfomicrobium*. Additionally, other studies, such as the one conducted by Kondo et al. [34], have validated the use of this specific primer pair. Therefore, based on that evidence, the primer pair DSR1F/RH3-dsr appeared suitable for our investigation.

Nevertheless, our findings exhibited inconsistencies when comparing low and high levels of SRB concentrations in both environmental and SR sludge samples (Figure 4). These discrepancies may arise from methodological limitations or be attributed to genetic, ecological, or functional characteristics of the SRB populations [3,16,29]. Thus, the evidence gathered does not support our initial approach of obtaining qPCR standards based on total and viable SRB counts, indicating the necessity of utilizing a conventionally accepted calibrator, such as a synthetic qPCR standard [57,58].

As a result, we conducted the qPCR analysis to determine the quantity of *dsrA* gene copies using the primers DSR1F/RH3-dsr-R and under-optimized reaction conditions (Figure 5). The standard curve fulfilled all the quality and acceptance criteria, establishing the validity of the analytical method. A qPCR efficiency theoretical value near 100% suggests that the polymerase enzyme operates at its optimal capacity [31,47,49]. Accordingly, to confirm its applicability to environmental and engineered sludge samples, we determined the number of *dsrA* gene copies in the above six samples. Compared with total SRB counts (Neubauer chamber), the number of copies of the target gene could be fewer, mainly due to bias associated with the microscopic method preventing accurate counting [16]. In terms of viable cell counts, as determined through culture plating, it was generally observed that they exceeded the number of *dsrA* gene copies, suggesting those exceeded viable counts are attributable to assimilatory SRB populations, which means they do not excrete sulfide. Culture techniques may not always accurately quantify complex bacterial communities due to factors such as media specificity or the inability of certain bacteria to grow under standard incubation conditions [16,59]. Our findings indicate that the time-consuming culture technique enables the cultivation of diverse SRB populations without distinguishing between assimilatory and dissimilatory SRB, whereas the *dsrA* gene specifically targets dissimilatory SRB [57,60]. Despite that, we observed a reasonable correlation between *dsrA* gene copies and viable SRB counts (Figure 6a), showing that culturing and qPCR analysis may be comparable methods for enumerating SRB. This trend is considered preliminary and subject to further investigation. Since the correlation coefficient is calculated based on means and standard deviations and is not resilient to outliers, we recommend considering a higher sample size in a wide range of SRB densities/concentrations. In a prior study examining microbial characterization of swelling bentonite clays, Vachon et al. observed that bacterial counts obtained through molecular techniques do not consistently align with the viable bacteria enumerated using culture-dependent methods [28].

The additional investigation should optimize the technical complexities associated with qPCR analysis compared to conventional techniques to enhance their performance. This study also aims to contribute to the establishment of a standardized approach for developing and validating qPCR assays for the enumeration of SBR in both engineered and environmental samples.

## 5. Conclusions

For the purpose of the present study, we assessed three techniques for the absolute enumeration of SRB in sludge samples: Neubauer chamber counting, culture plating, and qPCR analysis. Among the traditional methods, plate counting was the most effective, but it has issues as it is time consuming, labor intensive, and requires an anaerobic chamber. To overcome these limitations, our approach utilizing viable cells as normalized units met the acceptance criteria and provided evidence supporting enumeration through qPCR analysis. However, this method exhibited limitations when enumerating SRB in sludge samples with complex origins.

Afterward, we performed a qPCR analysis to normalize the SRB enumeration to *dsrA* gene copies, using a synthetic fragment (gBlock^®^). When comparing the two approaches, microscopic counting, once again, resulted in higher SRB counts. While the metabolically active populations enumerated through culture differed from the *dsrA* gene copies, a plausible positive correlation was observed. Hence, enumeration of the number of copies of a target gene proved valuable for obtaining accurate and reliable results enumerating dissimilatory SRB populations. Moreover, this approach has opened up new avenues of inquiry for future investigation.

## Figures and Tables

**Figure 1 biology-12-00985-f001:**
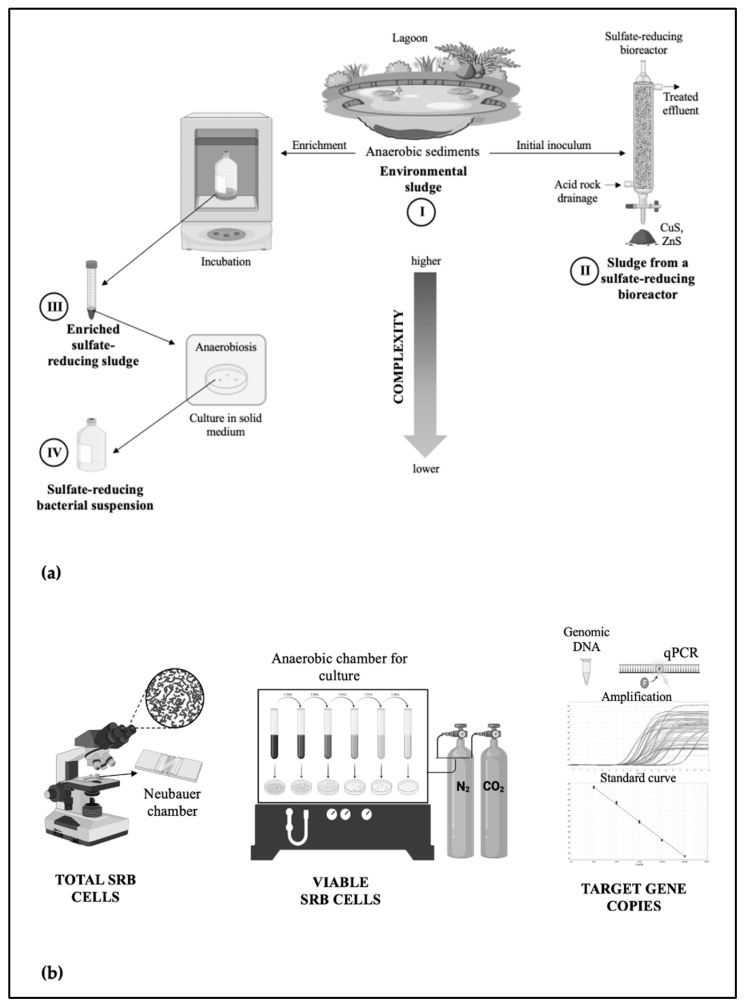
The schematic figure describes the methodology of this study. (**a**) Sources of sludge samples. (**b**) SRB enumeration techniques. It was created on BioRender.com on 21 June 2023.

**Figure 2 biology-12-00985-f002:**
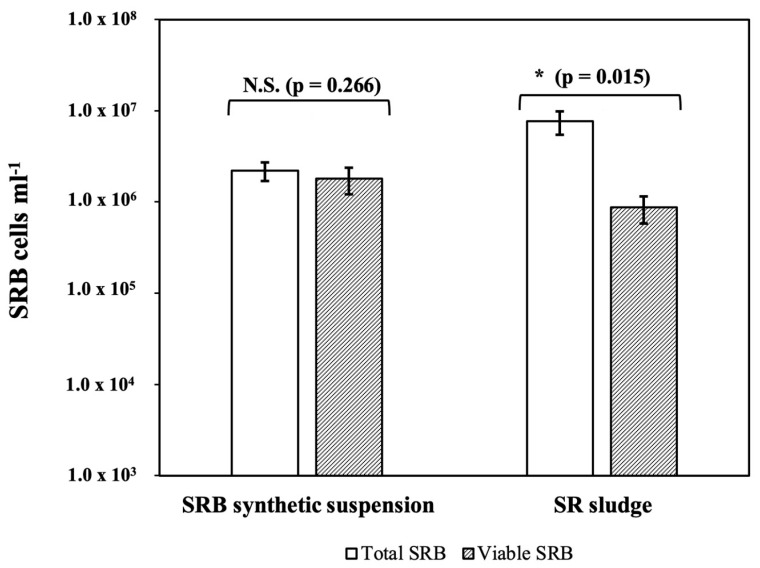
SRB density in a synthetic suspension and SR sludge, using the Neubauer chamber method (Total-open square) and by plating culture method (Viable-closed square) as conventional counting methods. Bars represent the average of measurements, and error bars represent the standard deviations of each mean. An asterisk (*) indicates a statistically significant difference according to the Student’s *t* test (*p* < 5%), and N.S. is not statistically significant. *p*-values are also indicated.

**Figure 3 biology-12-00985-f003:**
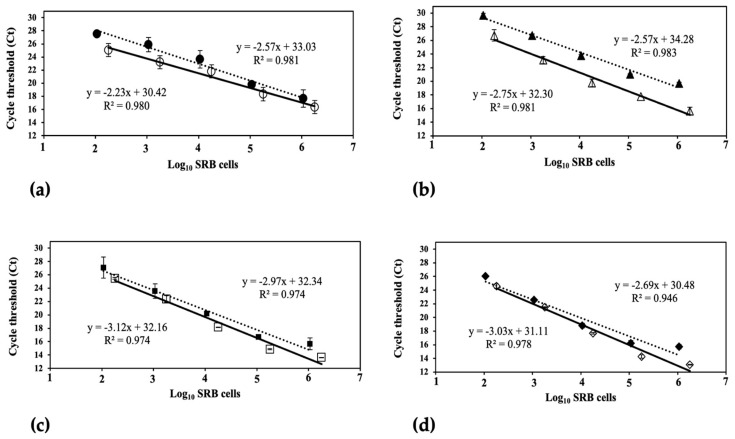
qPCR standard curves using a synthetic SRB suspension (solid line, open markers) and serial dilutions of SR enriched sludge (dotted line, filled markers) as calibrators. *dsrA* gene: (**a**) DSR1F/RH3-dsr-R (circles) and (**b**) RH1-dsr-F/RH3-dsr-R (triangles). *apsA* gene: (**c**) APS7-F/RH2-aps-R (squares) and (**d**) RH1-aps-F/RH2-aps-R (diamonds). Data points represent the average of three measurements, and the standard deviations are represented with error bars. According to ANOVA, there are no statistically significant differences between the slopes of each primer set (Appendix B, Table A2).

**Figure 4 biology-12-00985-f004:**
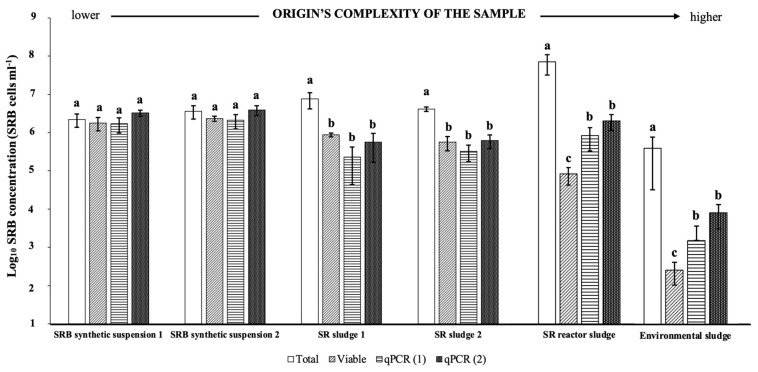
Quantification by qPCR using primers DSR1F and RH3-dsr-R and a mixture of synthetic SRB suspension (1) and serial dilutions of SR sludge (2) for standard curve plotting, Neubauer counting (Total) and plate culture (Viable) in six samples: two synthetic SRB suspensions, two SR sludge samples, a sludge from an SR bioreactor, and an environmental sludge. The bar plot represents the means for each level of a one-way analysis of variance (ANOVA). Error bars indicate the standard error of the mean. In each group (sample), bars sharing the same letter are not significantly different, according to Tukey’s test.

**Figure 5 biology-12-00985-f005:**
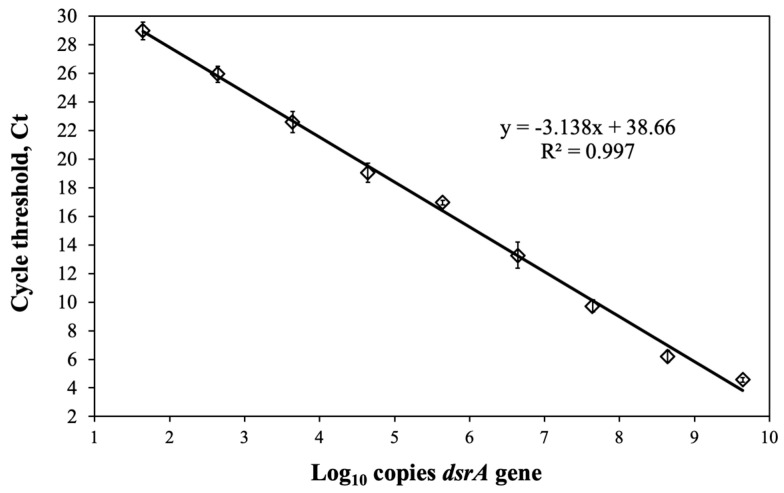
qPCR standard curves using serial dilutions of a double-stranded 222 bp gBlocks^®^ fragment of *dsrA* gene. Data points (open diamonds) represent the average of measurements, error bars, and standard deviations.

**Figure 6 biology-12-00985-f006:**
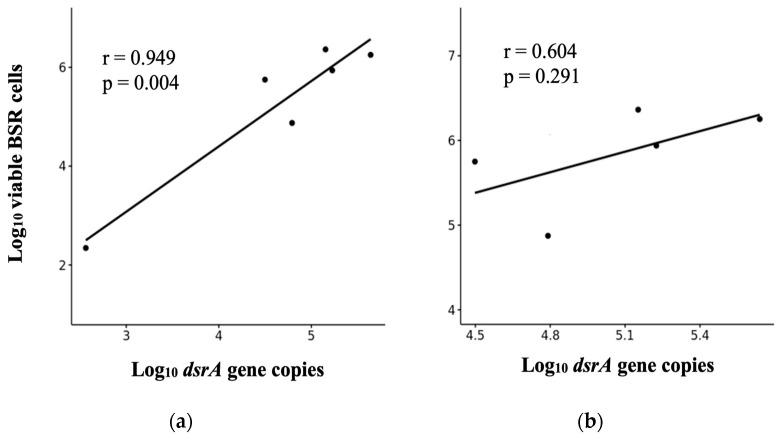
Scatter plots depicting the relationships between *dsrA* gene copies versus viable SRB cells, considering (**a**) all sludge samples, and (**b**) without the environmental sludge sample. The Pearson’s correlation coefficient is r (r > 0.7 indicates a strong relationship) and if the *p*-value is lower than 0.05, the correlation is considered statistically significant.

**Table 1 biology-12-00985-t001:** Quantification of SRB in four dilutions of a control standard (SRB suspension) using standard curves for sets of primers DSR1F/RH3-dsr-R and RH1-dsr-F/RH3-dsr-R for the *dsrA* gene, and APS7-F/RH2-aps-R and RH1-aps-F/RH2-aps-R for the *apsA* gene, obtained with a mixture of synthetic SRB suspension as calibrator.

Primer Set	SRB Density Determined by qPCR Standard Curve ^a^ and Percentage of Relative Error (%Ԑt) ^b^								
	Equation of a Standard Curve	Theoretical Value ^c^ (SRB Cells mL^−1^)							
		2.31 × 10^6^		2.31 × 10^5^		2.31 × 10^4^		2.31 × 10^3^	
		Cells mL^−1^	%Ԑ_t_	Cells mL^−1^	%Ԑ_t_	Cells mL^−1^	%Ԑ_t_	Cells mL^−1^	%Ԑ_t_
DSR1F/RH3-dsr-R	y = −2.23x + 30.42	2.10 × 10^6^	8.95	2.08 × 10^5^	9.89	2.46 × 10^4^	6.30	2.08 × 10^3^	9.89
RH1-dsr-F/RH3-dsr-R	y = −2.75x + 32.30	3.83 × 10^5^	98.3	1.26 × 10^4^	94.5	1.30 × 10^3^	94.4	1.07 × 10^2^	95.4
APS7-F/RH2-aps-R	y = −3.12x + 32.16	1.02 × 10^4^	55.8	3.09 × 10^5^	33.9	2.79 × 10^4^	20.9	1.28 × 10^3^	44.6
RH1-aps-F/RH2-aps-R	y = −3.03x + 31.11	8.88 × 10^4^	61.6	3.58 × 10^5^	54.8	2.65 × 10^4^	14.5	1.48 × 10^3^	35.9

^a^ qPCR standard curves were obtained using mixture of synthetic SRB suspension as calibrator. ^b^ Relative error (%Ԑt) was calculated as the absolute error (difference between the measured and theoretical or true values) divided into true values. ^c^ Theoretical or true SRB concentration was determined by plate culture.

**Table 2 biology-12-00985-t002:** Quantification of SRB in four dilutions of a control standard (SR enriched sludge) using standard curves for sets of primers DSR1F/RH3-dsr-R and RH1-dsr-F/RH3-dsr-R for the dsrA gene and APS7-F/RH2-aps-R and RH1-aps-F/RH2-aps-R for the *apsA* gene, obtained with serial dilutions of SR sludge coming from enriched subcultures as calibrator.

Primer Set	SRB Density Determined by qPCR Standard Curves ^a^ and Percentage of Relative Error (%Ԑt) ^b^								
	Equation of Standard Curve	Theoretical Value ^c^ (SRB Cells mL^−1^)							
		5.63 × 10^5^		5.63 × 10^4^		5.63 × 10^3^		5.63 × 10^2^	
		Cells mL^−1^	%Ԑ_t_	Cells mL^−1^	%Ԑ_t_	Cells mL^−1^	%Ԑ_t_	Cells mL^−1^	%Ԑ_t_
DSR1F/RH3-dsr-R	y = −2.57x + 33.26	6.27 × 10^5^	11.4	5.19 × 10^4^	7.73	4.84 × 10^3^	14.1	6.05 × 10^2^	7.45
RH1-dsr-F/RH3-dsr-R	y = −2.57x + 34.51	4.55 × 10^5^	19.3	2.28 × 10^4^	59.5	1.27 × 10^3^	125	8.63 × 10^2^	53.2
APS7-F/RH2-aps-R	y = −2.97x + 32.61	3.94 × 10^4^	93.0	7.73 × 10^3^	86.3	8.72 × 10^2^	84.5	1.65 × 10^2^	70.7
RH1-aps-F/RH2-aps-R	y = −2.69x + 30.72	1.05 × 10^4^	98.1	3.58 × 10^3^	93.6	1.34 × 10^2^	97.6	1.40 × 10	97.5

^a^ qPCR standard curves were obtained using SR sludge from subcultures as a calibrator. ^b^ Relative error (%Ԑt) was calculated as the absolute error (difference between the measured and theoretical or true values) divided into true values. ^c^ Theoretical or true SRB concentration was determined by plate culture.

**Table 3 biology-12-00985-t003:** Comparative average results of SRB quantification in six samples: two synthetic SRB suspensions, two SR sludge samples, an environmental sludge, and a sludge from an SR bioreactor.

Samples	Total Cells ^a^	Viable Cells ^b^	*dsrA* Copies ^c^	*dsrA* Gene Ratio	
				Copies:Total Cells	Copies:Viable Cells
SRB synthetic suspension 1	2.21 × 10^6^	1.79 × 10^6^	4.36 × 10^5^	0.20	0.24
SRB synthetic suspension 2	3.92 × 10^6^	2.31 × 10^6^	1.42 × 10^5^	0.04	0.06
SR sludge 1	7.63 × 10^6^	8.70 × 10^5^	1.68 × 10^5^	0.02	0.19
SR sludge 2	4.10 × 10^6^	5.63 × 10^5^	3.15 × 10^4^	0.01	0.06
Environmental sludge	2.51 × 10^5^	2.20 × 10^2^	3.66 × 10^2^	0.00 ^d^	1.66
SR reactor sludge	5.91 × 10^7^	7.48 × 10^4^	6.18 × 10^4^	0.00 ^d^	0.83

^a^ Value estimated through Neubauer chamber counting. ^b^ Value estimated through culture plating counting. ^c^ qPCR analysis using a synthetic *dsrA* gene fragment (gBlocks^®^). ^d^ The *dsrA* gene ratio is less than 0.01.

## Data Availability

There are no available datasets analyzed or generated during the study.

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
