# Peer review of "Comparative Methods for Quantification of Sulfate-Reducing Bacteria in Environmental and Engineered Sludge Samples"

_biology, 2023, doi:10.3390/biology12070985_

Round 1

Reviewer 1 Report

Reviewer comments

This paper reported Comparative Methods for Quantification of Sulfate-Reducing Bacteria in Environmental and Engineered Sludge Samples. After carefully review, some issues still need to be addressed before publication.

1. Innovation needs to be further improved;

2. The results of the method need to be compared, especially in terms of bias;

3. The research results need to be compared with relevant literature to highlight the advantages of the method;

4. The practical application of the method in Environmental engineering needs to be further supplemented.

 Minor editing of English language required

Author Response

Reviewer 1

This paper reported Comparative Methods for the Quantification of Sulfate-Reducing Bacteria in Environmental and Engineered Sludge Samples. After carefully review, some issues still need to be addressed before publication.

We appreciate the reviewer’s evaluations. Below, the changes are listed point-by-point. Based on your advice, we have thoroughly revised the original manuscript (all changes and additions are highlighted in blue).

  1. Innovation needs to be further improved;

The innovation statement was improved in the introduction section of the revised manuscript (lines 141 to 148). Based on state of the art, we provided a clear and brief explanation of how our study compares and validates existing enumeration techniques and proposes a method to quantify SRB in sludge (complex) samples.

  1. The results of the method need to be compared, especially in terms of bias;

Thank you for your comment. We ran additional statistical analyses and stated the limitations of our study in terms of bias, comparing them with other studies. While also we emphasize the strengths and contributions of our research. To see the changes, please refer to the revised manuscript's results, discussion and conclusions sections (highlighted in blue).

  1. The research results need to be compared with relevant literature to highlight the advantages of the method;

We have revised the cited bibliography regarding its relevance. As a result, we added 15 new references, including those to support the method's advantages. Please refer to the revised manuscript (lines 69-74, 84-86, 114-127).

  1. The practical application of the method in Environmental engineering needs to be further supplemented.

Thank you for your comment. In the revised manuscript, we have stated the extended practical application of the method in Environmental Engineering. Thus, we have focused on the enumeration of SRB in natural environments, bioreactors, and treatment systems as a crucial tool for assessing treatment efficiency, optimizing strategies, detecting system failures, and advancing bioremediation research by identifying metabolic capabilities with potential use. This information, supported by literature references, has been included in lines 67-74 of the revised manuscript.

Reviewer 2 Report

This study proved that a qPCR-based method optimized with dsrA gene copies as a calibrator. The author suggested it is a sensitive molecular tool for the absolute enumeration of SRB populations in engineered and environmental sludge samples. The quantification of functional microorganisms, such as SRB in this study, especially in environmental samples, is often necessary and difficult. Thus, I confirm the importance of this study. Overall, the manuscript is well structured and written. I recommend accepting this manuscript after revision. 

Major comments:

# How the primers of dsrA or apsA designed? What is the principle? How can the author confirm that the primers used in this study is qualified for all SRB? Please clarified. 

# Line 480, The author state: “Therefore, in this study, bacterial counts for qPCR standard curves were based on values obtained from colony counts by culture plating (viable).” I wonder how the author qualitative identification of a SRB bacterium by culture plating? The author said in Line 179 “Bacteria morphology was determined since SRB cells' characteristic black or dark brown color [21]”, which seems that the author identified the SRB through the color change of the colony. If this is the case, what is the standard of the color (black or dark brown) of the colony?

Minor comments:

#Line 313-317 should incorporate into Figure 2.

#All the Tables should have the corresponding statistical analysis

#Line 447 6/5/2023 10:35:00 AM?

#Line 564 “In terms of viable cell counts, as determined through culture plating, it was generally observed that they exceeded the number of dsrA gene copies, suggesting SRB populations are mainly assimilatory, which means they do not excrete sulfide.” Please explain the detail about why the SRB are mainly assimilatory. 

# What is the means of “COD”? Line 143, 162…

Author Response

Reviewer 2

This study proved that a qPCR-based method optimized with dsrA gene copies as a calibrator. The author suggested it is a sensitive molecular tool for the absolute enumeration of SRB populations in engineered and environmental sludge samples. The quantification of functional microorganisms, such as SRB in this study, especially in environmental samples, is often necessary and difficult. Thus, I confirm the importance of this study. Overall, the manuscript is well structured and written. I recommend accepting this manuscript after revision. 

Thank you for your detailed evaluations and suggestions. The original manuscript has been revised thoroughly according to your advice.

Major comments:

# How the primers of dsrA or apsA designed? What is the principle? How can the author confirm that the primers used in this study is qualified for all SRB? Please clarified. 

Ben-Dov and coworkers designed these primers from conserved regions of multiple alignments of dsrA or apsA sequences obtained from constructed libraries of SRB in industrial wastewater ponds (lines 252-254 in the revised manuscript).

According to the same study (Ben-Dov et al, 2007), the primers DSR1F/RH3-dsr-R displayed high specificity for most SRB species, including members of the genera Desulfovibrio, Desulfomonas, Desulfatibacilus, Desulfomicrobium, Desulfobacterium, Desulfosarcina, Desulfonema, Desulfofaba, Desulfomusa, Desulfohalobiaceae, Desulfotignum, Desulfomaculum, Desulfofacinum, Desulfonatronum, Desulfarculus, Desulfovirga, Desulfobulbus, Desulfobacter, Desulfococcus, Desulfonema, Desulfocapsa, Desulfofustis, Desulfobacula, Desulfocella and Desulfofospira. In our previous research (Zambrano et al., 2021), V3 and V4 regions of 16S rRNA genes were sequenced in the enriched sulfate-reducing sludge and the sludge from a sulfate-reducing bioreactor, being taxonomically identified: Desulfobacteraceae, Desulfuromonadales, Desulfotomaculum, Desulfovibrio, Desulfobulbaceae, Desulfobacca, Desulfocurvus, Desulfoluna, Desulfovibrionales, and Desulfomicrobium. Moreover, other studies have validated de use of this pair of primers (Kondo et al, 2004). Therefore, the primers DSR1F/RH3-dsr used in our study seemed to be appropriate.

Thank you for your suggestion. The explanatory text has been included in the revised version of the manuscript (lines 256-258 and 555-568).

# Line 480, The author state: “Therefore, in this study, bacterial counts for qPCR standard curves were based on values obtained from colony counts by culture plating (viable).” I wonder how the author qualitative identification of a SRB bacterium by culture plating? The author said in Line 179 “Bacteria morphology was determined since SRB cells' characteristic black or dark brown color [21]”, which seems that the author identified the SRB through the color change of the colony. If this is the case, what is the standard of the color (black or dark brown) of the colony?

Thank you for your question. In the culture plating to enumerate the viable SRB, all colony-forming units (CFU) are counted (lines 212-224 in the revised manuscript). Since the culture medium just provides sulfate as an electron acceptor and the atmosphere during the incubation was a mixture of nitrogen and carbon dioxide (N2/CO2, 80/20 v/v), all grown microorganisms were sulfate reducers. So, the used culture medium was selective, not differential and changing color is not applicable.

On the other hand, for the SRB enumeration (total cells) using the Neubauer chamber just where counted the black or dark brown color cells (lines 203-211 in the revised manuscript).

Minor comments:

#Line 313-317 should incorporate into Figure 2.

Thank you for your comment. Corrected in the legend of that figure (Figure 3 and lines 353-355 in the revised manuscript).

#All the Tables should have the corresponding statistical analysis

Thank. This change was done when it was pertinent in tables and figures in the revised manuscript.

#Line 447 6/5/2023 10:35:00 AM?

Thank you. We have eliminated that line in the revised manuscript.

#Line 564 “In terms of viable cell counts, as determined through culture plating, it was generally observed that they exceeded the number of dsrA gene copies, suggesting SRB populations are mainly assimilatory, which means they do not excrete sulfide.” Please explain the detail about why the SRB are mainly assimilatory. 

Thank you for your review. SRB are a group of microorganisms with the unique metabolic capability to use sulfate (SO42-) as an electron acceptor during anaerobic respiration. They play a significant role in the sulfur cycle by reducing sulfate to sulfide (H2S) as a by-product. SRB could use sulfide to synthesize sulfur amino acids (methionine and cysteine) in the assimilatory pathway.  If sulfide is excreted, the pathway is dissimilatory.  The dsrA gene codifies the alpha subunit of the dissimilatory sulfite reductase and is a specific marker for dissimilatory SRB (lines 96-105 of the revised manuscript).

Since the viable cell counts exceeded the number of dsrA gene copies (lines 584-586 in the revised manuscript), it is possible to infer that those excess counts are attributable to assimilatory SRB populations. We rewrote lines 586-587 in the revised manuscript for more clarity in the text.

# What is the means of “COD”? Line 143, 162…

Thank you for your comment. Yes, we had not indicated the meaning of COD in the entire text. COD is the Chemical Oxygen Demand, an indirect measure of the substrate and electron donor concentration. We have corrected this point in the revised manuscript where the COD is first mentioned (line 175 in the revised manuscript), as follows:

… with sodium acetate equivalent to 2.5 g of Chemical Oxygen Demand (COD) l−1 as organic substrate …

Reviewer 3 Report

In the manuscript "Comparative Methods for Quantification of Sulfate-Reducing Bacteria in Environmental and Engineered Sludge Samples" by Aracely Zambrano-Romero and collaborators, the authors studied different techniques with what to study sulfate-reducing bacteria in sludge samples. Although the approach is interesting, there might be several flaws to have fully sure about the results. Many of the provided results do not have that strong evidence to confirm the ideas (see concerns at the end)

The Simple Summary's one sentence needs more meaning: "However, it does not work for complex samples." What is meant by complex samples?

In the case of the methods, it is not easy to read and understand. A schematic figure describing the methodology would be very beneficial, especially related to the cultivation, qPCR, etc.

Figure 1 – If the standard deviation is shown and it is not at all overlapping in the case of the SR sludge, it is doubtful that the significant difference is only p<0.05.

How was it validated that this or that method was better? How do we really know if the abundances of organisms or genes are over- or underestimated as any of these methods (Neubauer chamber (total count), culture plating (viable count, CFU), and qPCR) is perfect?

Figure 5 – What are p-values?

Figure 5 (b) and (c) – This relationship is not proving anything, as only one point (maybe an outlier) creates the relationship between variables.

"This methodology has not yet been deeply evaluated to analyze complex sludge samples." – Based on what, we can say that these sludge samples were complex? Complex to compare with what?

Main issues:

Complex samples – What defines complex samples if they are not compared to other types of samples?

Which of the following methods can we use to estimate if some genes/organisms are over- or underestimated?

Results seem to be biased, e.g., check 5c.

Why not normalize with dry matter as it is usually done?

Minor editing of English language required

Author Response

Reviewer 3

In the manuscript "Comparative Methods for Quantification of Sulfate-Reducing Bacteria in Environmental and Engineered Sludge Samples" by Aracely Zambrano-Romero and collaborators, the authors studied different techniques with what to study sulfate-reducing bacteria in sludge samples. Although the approach is interesting, there might be several flaws to have fully sure about the results. Many of the provided results do not have that strong evidence to confirm the ideas (see concerns at the end)

We highly value the reviewer's assessments. Here, we present the modifications made in response to each point raised. The original manuscript has undergone a comprehensive revision based on your valuable recommendations.

The Simple Summary's one sentence needs more meaning: "However, it does not work for complex samples." What is meant by complex samples?

The suggested change has been made accordingly (lines 27-28 in the revised manuscript). In the revised manuscript, we rewrote that sentence in the Simple Summary section as follows:

… However, it does not work for all samples due to their complex physical, chemical, and microbiological traits.

In the case of the methods, it is not easy to read and understand. A schematic figure describing the methodology would be very beneficial, especially related to the cultivation, qPCR, etc.

Thank you for raising this point. In the revised manuscript, we have included a schematic figure describing the methodology (Fig. 1a and 1b). As a result, the other figures were renumbered, and the changes are highlighted in the revised manuscript.

Figure 1 – If the standard deviation is shown and it is not at all overlapping in the case of the SR sludge, it is doubtful that the significant difference is only p<0.05.

Thanks for the suggestion. The p-value was changed to the specific value, not just the standard threshold, which makes the message of the figure more solid (Fig. 2 in the revised manuscript). There are statistically significant differences between SRB total cells and SRB viable cells in the SR sludge (p = 0.015). The calculated p-values were also included in the text (lines 312-313 and 319 of the revised manuscript).

How was it validated that this or that method was better? How do we really know if the abundances of organisms or genes are over- or underestimated as any of these methods (Neubauer chamber (total count), culture plating (viable count, CFU), and qPCR) is perfect?

Thank you for your comment. Well, no analytical method is perfect. Factors such as accuracy, sensitivity, repeatability, and specificity, are essential considerations when evaluating the performance and reliability of analytical methods. We established the number of SRB viable cells as the theoretical true valor because they are the live cells in culture. Culture-dependent techniques are not perfect (cultivation methods may fail to account for viable yet non-culturable bacteria and slow-growing microorganisms). We initially normalize the qPCR units to SRB viable cells, using genomic DNA from diverse SRB species as standard.  Nevertheless, the evidence gathered did not support this initial approach.  Therefore, we used a synthetic qPCR standard (gBlocks), in which normalized units are dsrA gene copies. Then, we sought a correlation between total cells and viable cells, total cells and dsrA gene copies, and viable cells and dsrA gene copies. Lastly, both culture plating and qPCR are regarded as reliable analytical methods for quantifying SRB in the context of using the bioprecipitation of metals as a bioremediation approach.

Figure 5 – What are p-values?

Thank you for your comment. We included the p-values in that figure (Fig. 6 in the revised manuscript).

Figure 5 (b) and (c) – This relationship is not proving anything, as only one point (maybe an outlier) creates the relationship between variables.

Thanks for the suggestion. After checking that correlation is driven from one point, we decided to modify that figure (Figure 6 in the revised manuscript) and change the discussion and conclusion sections. Since either total with viable count cells or total count cells with dsrA copy number measured by gBlocks are not correlated, those figures were removed to Appendix C (Figures C1a and C1b).

The changes were made in the revised manuscript in lines 445-460, figure 6 and 593-602.

"This methodology has not yet been deeply evaluated to analyze complex sludge samples." – Based on what, we can say that these sludge samples were complex? Complex to compare with what?

Thank you. The changes were done in lines 462-466 in the revised manuscript.

Main issues:

Complex samples – What defines complex samples if they are not compared to other types of samples?

We agree with the suggestion to explain what complex samples are. Briefly, sludge and soil samples from engineered or natural environments are complex in their physical, chemical, and microbiological traits. They have diverse physical properties and consist of organic and inorganic compounds and pollutants. These samples also host a variety of microorganisms, creating intricate microbial communities with varying compositions and metabolic activities. Analyzing specific groups of microorganisms, such as SRB, presents challenges due to diverse environments.

The explanatory text has been included in the revised version of the manuscript (lines 114-127).

Which of the following methods can we use to estimate if some genes/organisms are over- or underestimated?

Based on our findings, both the qPCR method and culture plating can be utilized while considering the respective advantages and limitations of each approach. Please, refer to the Conclusions section (lines 593-596 in the revised manuscript).

Results seem to be biased, e.g., check 5c.

After checking the results, problems were only identified in Figure 5c (Fig 6 in the revised manuscript) because the counts from the environmental sludge drove the correlation. Corrections in the text were made to reflect this (lines 445-460, figure 6 and 593-602, in the revised manuscript). Additionally, supplementary figure C1, is added to show the lack of correlation between the two conventional enumeration techniques and total cells with qPCR.

Why not normalize with dry matter as it is usually done?

We sincerely apologize, but we are currently unable to understand your question. We kindly request that you provide further details or clarification so we can answer it.

Round 2

Reviewer 1 Report

Accept in present form

Reviewer 2 Report

I have no further comment.